# An Uncommon Case of Moyamoya Syndrome Is Accompanied by an Arteriovenous Malformation with the Involvement of Dural Arteries

**DOI:** 10.3390/ijms24065911

**Published:** 2023-03-21

**Authors:** Chingiz Nurimanov, Iroda Mammadinova, Yerbol Makhambetov, Serik Akshulakov

**Affiliations:** Vascular and Functional Neurosurgery Department, National Center for Neurosurgery, Astana 010000, Kazakhstan

**Keywords:** arteriovenous malformation, arteriovenous fistula, moyamoya syndrome, surgical management

## Abstract

(1) Background: This report describes the surgical management of a case of concurrent AVM with the involvement of dural arteries and moyamoya syndrome. Given the infrequency of this combination, there is currently no established management strategy available. (2) Case Description: A 49-year-old male patient with multiple symptoms including headaches, tinnitus, and visual impairment diagnosed with the coexistence of an arteriovenous malformation with the involvement of dural arteries and moyamoya syndrome was admitted to the national tertiary hospital. The patient underwent surgical management through embolization of the AVM from the afferents of the dural arteries, which has resulted in positive clinical outcomes. However, this approach may not be suitable for all cases, and a multidisciplinary team approach may be required to develop an individualized treatment strategy. (3) Conclusion: The contradictory nature of the treatment approaches in cases of combined AVM with the involvement of dural arteries and MMD highlights the complex nature of this condition and the need for further research to identify the most effective treatment strategies.

## 1. Introduction

The combination of arteriovenous malformation (AVM) with moyamoya syndrome and dural involvement of afferent arteries is an exceptionally rare vascular pathology with severe complications and potentially fatal outcomes [1]. Arterial transdural blood supply is rarely seen in cerebral arteriovenous malformations (AVMs) where the blood supply to the lesion is derived from the branches of the external carotid artery (ECA) traversing through the dura mater. This is an atypical feature in AVMs, particularly when the ECA supplies the moyamoya-pattern territory. Recent publications have reported a prevalence of approximately 7% for cerebral arteriovenous malformations (AVMs) that involve dural arteries [2,3]. Moyamoya disease (MMD) is an idiopathic steno-occlusive condition that primarily affects the anterior circulation, while MMS, also known as the “moyamoya” pattern or phenomenon, is a chronic cerebrovascular condition that causes aberrant collateral vascular networks to develop along with internal carotid artery stenosis or occlusion. Moyamoya syndrome (MMS) is associated with a variety of underlying conditions. These may include atherosclerosis, radiation-induced vasculopathy, fibromuscular dysplasia, neurofibromatosis type 1, Graves disease, Down syndrome, neuro infections, sickle cell anemia, and in rare cases, systemic lupus erythematosus, Turner syndrome, Noonan syndrome, and Alagille syndrome [4,5]. Other vascular abnormalities, such as cerebral aneurysms, can occasionally occur in tandem with MMD. The incidence of intracranial aneurysms in patients with MMD is relatively high, ranging from 3.4% to 14.8%. The underlying mechanisms are not yet fully understood, but it is believed to be caused by the high-pressure flow via relatively small collaterals and vascular endothelial congenital abnormalities [1,6]. The coexistence of a brain arteriovenous malformation (AVM) with MMD is relatively rare (1.7 per 1000 persons).

Managing arteriovenous malformation involving dural arteries, in addition to moyamoya syndrome, in patients presenting with non-hemorrhagic neurological deficits such as visual impairment and pulsatile tinnitus is a challenging and controversial task.

High-flow AVMs with dural components can become symptomatic and cause secondary ischemic neurological deficits due to chronic venous hypertension and compromised vascular flight in adjacent tissues. This can lead to progressive or abrupt focal deficits, seizures, cognitive decline, and other neurological symptoms that can mimic dementias or neurodegenerative diseases [7].

Moyamoya disease (MMD) is believed to be caused by a combination of angiogenic factors and hemodynamic stress that leads to the development of arteriovenous malformations (AVMs) [1,8,9]. While there is typically no connection between MMD and dural arteriovenous fistulas (dAVFs), there have been uncommon cases of a unilateral "moyamoya" pattern that is associated with an ipsilateral dAVF [10,11]. However, the occurrence of MMD in conjunction with AVMs involving dural arteries has not been reported in the literature, making treatment of these combined conditions particularly challenging. This report aims to address the difficulties in managing cases of MMD that coexist with AVMs involving dural arteries.

## 2. Case Presentation

A 49-year-old right-handed man was admitted to the hospital with a history of headaches and blurred vision that had persisted for three months. Approximately one year prior, he had experienced transient ischemic attacks that caused temporary motor weakness in his right arm and leg. Upon admission, a neurological examination did not reveal any neurological deficits, except for visual impairment and tinnitus. The patient’s visual acuity (thereafter–VA) was 4/10 in the right eye and 10/10 in the left eye.

### 2.1. Diagnostic Procedures

An MRI of the brain revealed a 35 × 33 × 30 mm nidus of malformed vessels with an enlarged draining vein in the left occipital region, primarily draining into the superior sagittal sinus (Figure 1A–F).

In addition, MR angiography demonstrated restricted blood flow in the cavernous segment of the patient’s left internal carotid artery (ICA), with occlusion observed in the supraclinoidal segment, followed by the reconstitution of blood flow in the distal branches (Figure 2A–F).

The cerebral angiography (Figure 3A–D) reveals an AVM (Spetzler–Martin Grading Scale Grade III) with the involvement of dural arteries in the left occipital lobe and a left-side “moyamoya” pattern. Many small afferents are noted from the left external carotid artery (occipital, superficial temporal, and middle meningeal arteries), from the right external carotid artery (occipital, superficial temporal), and from the left middle cerebral artery (posterior parietal artery). The AVM drained into enlarged cortical veins that emptied into the superior sagittal sinus. The “moyamoya” pattern was characterized by the occlusion of the internal carotid artery on the left side at the ophthalmic segment and a decrease in the number and size of the cortical branches of the left anterior and medial cerebral arteries. 

### 2.2. Treatment

Because of the small caliber and tortuosity of the afferents from the intracranial vessels and the extended length of the left middle meningeal artery that supplied the territory of the left MCA, an endovascular embolization of the AVM with a liquid embolic agent was performed via left-side occipital craniotomy closer to the AVM and a direct afferent (middle meningeal artery) puncture (Figure 4 and Figure 5A–C).

After the surgical procedure, the patient’s symptoms gradually improved during the postoperative period. The visual impairment, which was initially at VA 4/10, improved to 8/10 in the right eye, and the tinnitus also regressed. After a week of hospitalization, the patient was discharged for dynamic monitoring to ensure continued improvement. 

The cerebral angiography performed 10 months after surgery demonstrated embolized superior, inferior, and lateral parts of the AVM. Many small afferents are seen from the occipital, superficial temporal, and meningeal arteries from both sides, as well as cortical vein drainage to the superior sagittal sinus (Figure 6 and Figure 7).

## 3. Discussion

This case is a rare combination of three cerebral lesions. The primary goal of treatment was to determine the necessity of intervention and select the most appropriate treatment options for each pathology. It is essential to note that MMD and AVM with the involvement of dural arteries are two distinct forms of cerebral lesions with unique pathological processes. The incidence rate of AVM combined with MMD is relatively low at 1.7 per 1000 people, which is ten times lower than the incidence rate of each condition separately in the general population. The prevalence of AVM with involvement of dural arteries linked with MMD is uncertain, with only two previously reported cases of concomitant MMD and dAVF and only two patients characterized as having concurrent MMS and dAVF [5,9]. According to the research, angiogenesis is the mechanism of de novo AVM formation in MMD patients, particularly after stereotactic radiosurgery [9,11,12]. However, in this particular case, the patient had no prior history of radiosurgery, which makes it unique. It is worth noting that the absence of a history of radiosurgery does not rule out the possibility of angiogenesis contributing to the formation of the AVM. Therefore, a thorough evaluation of the patient’s medical history and imaging studies is crucial to determine the most appropriate treatment approach. This case highlights the importance of considering all potential underlying mechanisms when evaluating and treating patients with cerebral lesions.

In this case, moyamoya syndrome, in combination with an arteriovenous malformation fed by dural arteries, can present with an acute course of symptoms such as pulsatile tinnitus and visual disturbance. Symptoms and signs may present with periods of remission and relapse due to changes in venous drainage.

Pulsatile tinnitus can be considered a sign of a serious and potentially life-threatening condition [13] and cause discomfort for the patient. As has been reported in a study by Yong-Hwi An et al., among patients with dural arteriovenous fistula presenting with pulsatile tinnitus that underwent endovascular embolization, 24 patients (92.3%) reported a decrease in the manifestation of pulsatile tinnitus, and 21 patients (80.8%) achieved complete resolution of symptoms [14]. This highlights the importance of timely and accurate diagnosis and appropriate treatment of a combined disease.

Managing arteriovenous malformation involving dural arteries, in addition to moyamoya syndrome, in patients presenting with non-hemorrhagic neurological deficits such as visual impairment due to vascular steal phenomenon is a challenging and controversial task. High-flow AVMs with dural components may become symptomatic and cause ischemic neurological deficits secondary to chronic venous hypertension and vascular steal from adjacent compromised tissue. The symptoms may present as progressive or abrupt focal deficits, seizures, and cognitive decline, which can resemble dementias or neurodegenerative diseases [7].

Palliative embolization is performed when radiosurgery is not suitable. It is used in cases where long-standing venous hypertension or vascular steal has resulted in ischemic neurological deterioration and morbidity. The goal of palliative embolization is to alleviate symptoms caused by vascular steal syndrome, improve quality of life, and reduce the risk of further complications [15]. It is important to consider a case-by-case clinical presentation, imaging findings, and comorbidities when selecting a treatment option.

Hemodynamic stress explains the process through which AVM initially develops and is followed by MMD. According to the limited research, an ischemia environment and subsequent angiogenesis have been postulated to play crucial roles in the development of AVFs. High blood flow to the AVM causes turbulent flow in the ICA, affecting the endothelial protection mechanisms and leading to gradual occlusion of the afflicted ICA. MMD collateral vessels form in order to protect brain tissue from hypoxia [10,16].

The concepts of treating AVM in conjunction with MMD are clearly contradictory, which is the most fascinating aspect of this condition. The challenge of choosing the optimal sequence of surgical treatment for cases that involve both AVM and MMD, along with dural afferents, remains an unresolved issue with conflicting arguments in the literature.

In cases where an arteriovenous malformation (AVM) is combined with moyamoya disease (MMD), treatment recommendations are based on the patient’s symptoms and cerebral perfusion status. In such cases, bypass surgery for MMD is recommended. However, revascularization interventions, which improve blood flow to the brain, may lead to the growth of the AVM nidus due to the collateral blood vessels [16,17]. Therefore, the decision to perform bypass surgery in such cases must be considered carefully, taking into account the patient’s individual condition and the risks associated with the different treatment options. Bypass surgery may assist in preventing further growth of the AVM nidus and reduce the risk of associated complications, but it is not always the best option for every patient. When symptoms of AVM appear in patients with combined MMD, surgical excision of the AVM is often necessary. However, there is a risk that the critical collateral connections to nearby ischemic brain tissue may be damaged during the surgery. In such cases, gamma knife radiosurgery is an effective therapy option for AVM.

Nevertheless, it is important to note that in rare cases, gamma knife radiosurgery may initiate MMS and worsen the patient’s condition. Therefore, treatment decisions must be made on a case-by-case basis, considering the risks and benefits associated with each option. Additionally, close follow-up after any intervention is necessary to control the patient’s progress and make adjustments to the treatment plan as needed [1,9]. In this case, after selective embolization of the meningeal artery, which was identified from angiography as the feeding artery for the AVM, the patient reported a significant improvement in his tinnitus and visual field defect. This highlights the effectiveness of embolization as a treatment option for AVM-related symptoms.

Embolization is a potential treatment option for patients with AVM and AVF, especially when there is an increasing steal effect as was observed in the present case. Lo Presti et al. reported the successful treatment of a case of AVF with MMD using Onyx transarterial embolization [17]. However, complete embolization of AVM and AVF may not be justified in cases where the blood supply to the ACA and MCA may depend on the AVM, as embolization of afferent vessels could compromise the collaterals. In such scenarios, the evidence to support surgical treatment over medical therapy is not conclusive.

## 4. Conclusions

The contradictory nature of treatment approaches in cases of combined AVM involving dural arteries and MMD highlights the complex nature of this condition and the need for further research to identify the most effective treatment strategies. Surgery may be considered as a treatment option for patients with MMS and associated AVM or AVF depending on the patient’s clinical characteristics and cerebral hemodynamic condition.

## Figures and Tables

**Figure 1 ijms-24-05911-f001:**
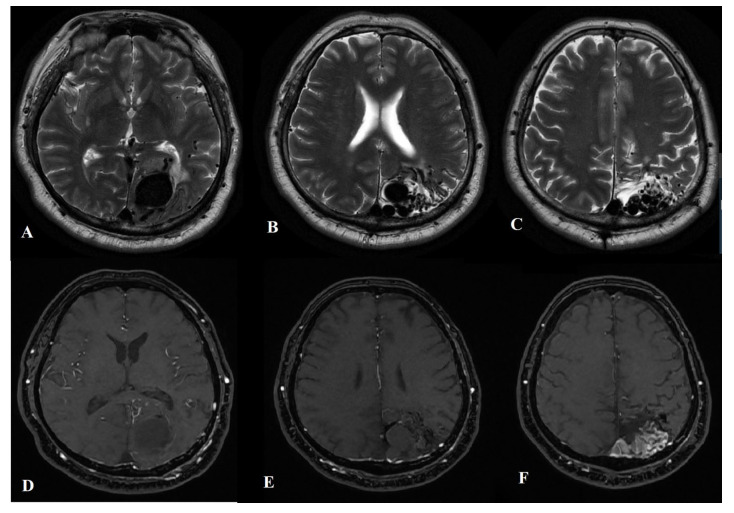
Axial T2-weighted (**A**–**C**) brain MRI and TOF MRA (**D**–**E**) demonstrating the left occipital AVM with an enlarged draining vein.

**Figure 2 ijms-24-05911-f002:**
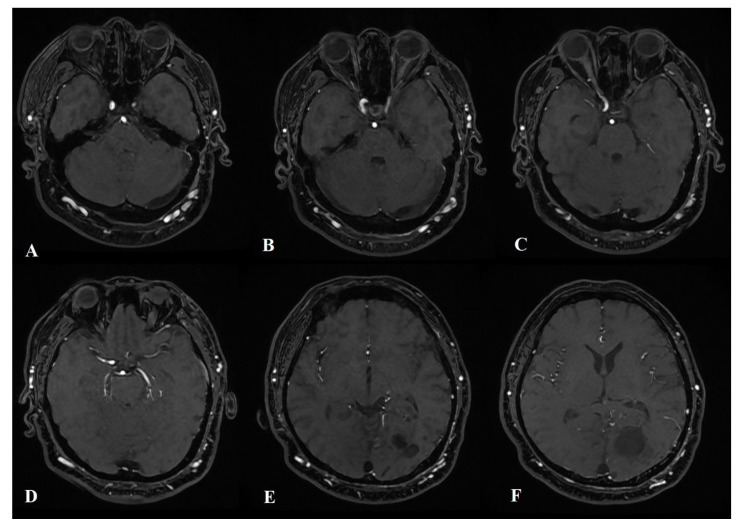
TOF MRA demonstrating the narrowing of the left ICA in the cavernous segment (**A**,**B**), its occlusion in the supraclinoid segment (**C**), and the reconstitution of flow in the distal branches (**D**–**F**).

**Figure 3 ijms-24-05911-f003:**
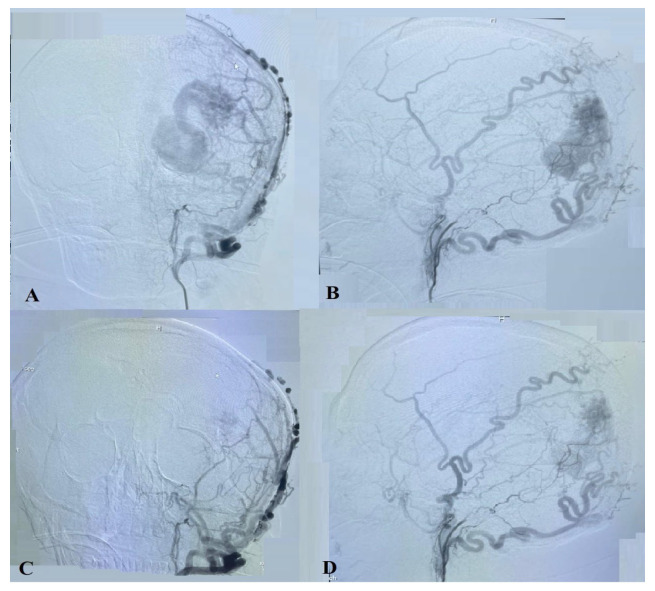
A carotid angiogram from left ICA (**A**–**D**) shows a left occipital AVM fed by many small arteries (left occipital, left superficial temporal, and left middle meningeal arteries).

**Figure 4 ijms-24-05911-f004:**
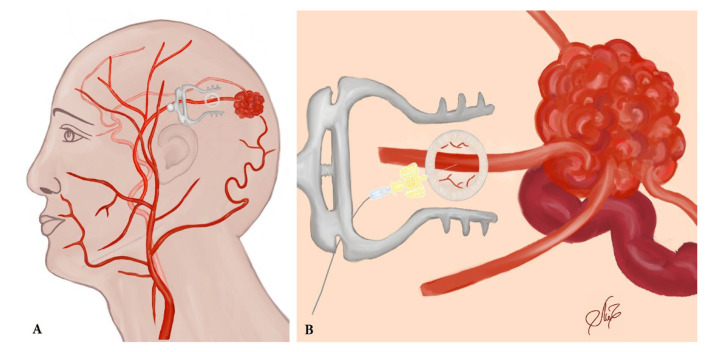
(**A**) An illustration of the arteriovenous malformation (AVM) with its afferent connections and the occipital craniotomy. (**B**) An illustration is provided to demonstrate the afferent puncture of the middle meningeal artery through the occipital craniotomy.

**Figure 5 ijms-24-05911-f005:**
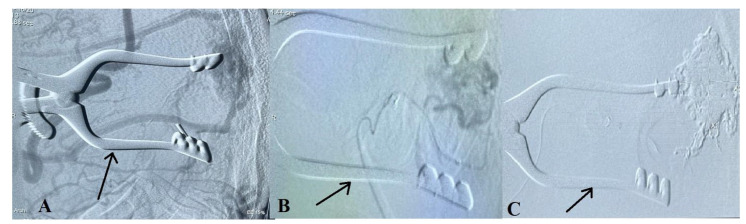
A carotid angiogram (**A**) after occipital craniotomy showing the AVM. Superselective angiography image from middle meningeal artery afferent (**B**). Angiogram view after embolization (**C**). Jansen retractor marked by the arrow.

**Figure 6 ijms-24-05911-f006:**
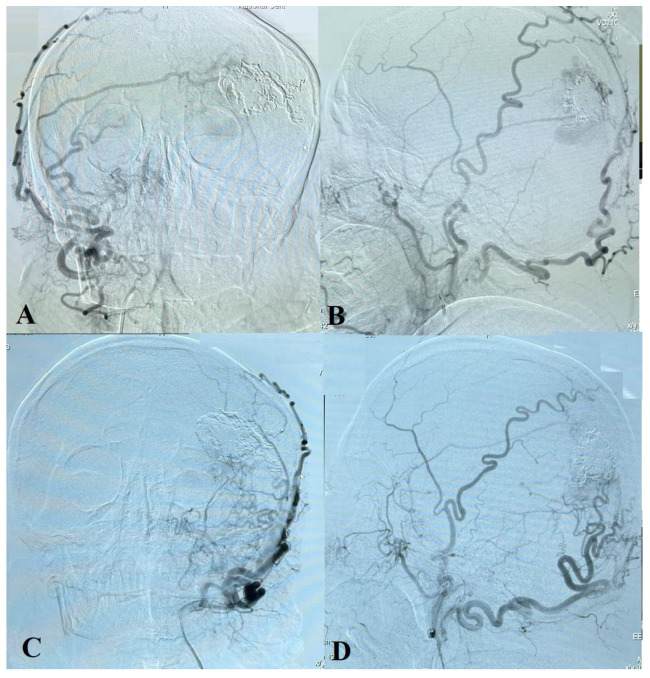
Carotid angiogram from the right ICA (**A**,**B**) and from the left ICA (**C**,**D**) demonstrates the AVM’s embolized superior, inferior, and lateral parts. Many small afferents are seen from the occipital, superficial temporal, and meningeal arteries.

**Figure 7 ijms-24-05911-f007:**
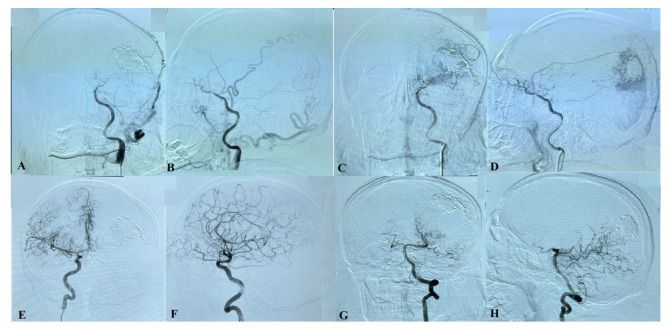
A carotid angiogram from the left ICA (**A**–**D**) demonstrates occlusion of the left ICA at the level of the ophthalmic segment. A cerebral angiogram from the right ICA (**E**,**F**) and the left vertebral artery (**G**,**H**).

## Data Availability

Not applicable.

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
