# Peer review of "An Uncommon Case of Moyamoya Syndrome Is Accompanied by an Arteriovenous Malformation with the Involvement of Dural Arteries"

_ijms, 2023, doi:10.3390/ijms24065911_

Round 1

Reviewer 1 Report

This is an interesting and well written case report presentation that fits the Journal scope. The paper aimed to address the difficulties in managing cases of MMD that coexist with AVMs involving dural arteries.

Overall, the paper is well written. I have no doubt that the article can be accepted in the current form. I want to congratulate the authors for their work.

Author Response

Dear reviewer,

Thank you for taking the time to review our article "An uncommon case of Moyamoya syndrome accompanied by an arteriovenous malformation with the involvement of dural arteries". We appreciate your positive feedback and are pleased to hear that you found our article interesting.

We believe that our findings can contribute to a better understanding of the etiology, clinical presentation, and management of these rare conditions.

Thank you once again for your unwavering support and for taking the time to review our article. We are grateful for your invaluable feedback, and we look forward to further cooperation.

Reviewer 2 Report

Manuscript ID: ijms-2278738

Type of manuscript: Case Report

Title:  An uncommon case of Moyamoya syndrome is accompanied by an arteriovenous malformation with the involvement of dural arteries.

1)     Introduction.

This article describes the case of a 49-year-old man “ with a history of head-aches and blurred vision that had persisted for three months. Approximately one year prior, he had experienced transient ischemic attacks that caused temporary motor weakness in his right arm and leg. Upon admission, a neurological examination did not reveal any neurological deficits, except for visual impairment and tinnitus. The patient's visual acuity (thereafter – VA) was 4/10 in the right eye and 10/10 in the left eye. An MRI of the brain revealed a 35x33x30mm nidus of malformed vessels with an enlarged draining vein in the left occipital region, primarily draining into the superior sagittal sinus”. This is a very interesting and well written presentation.

2) General Comment:

However, it is difficult to draw conclusions on a very specific case. The interest is therefore to be pondered.

Moreover, the reading would be more pleasant if the document was more structured (see an example below)

1.Introduction

2. Materials and Methods

2.1. Patient

2.2. Evaluations

2.3. Imaging and Pathological Diagnosis

2.4. Treatment

3. Discussion

…..

3) Conclusion.

Due to various comments (especially on the structure of this paper), this article can be easily improved before to be published in IJMS

I would like to thank the authors for writing this article, and the editor for inviting me to review the article. It was a very interesting task.

Author Response

Dear reviewer,

Point 1: Thank you for taking the time to review our article and for your feedback on the specific case study presented in our work. We appreciate your thoughts on the need to carefully consider the implications of the study's findings.

We understand that the case study presented in this article may not be generalizable to all cases, and the limitations of the study should be considered. However, we believe that this case study contributes to the understanding of an uncommon manifestation of Moyamoya syndrome and highlights the need for further research in this area.

We also agree that it is essential to consider the broader implications of this case study in the context of existing research on Moyamoya syndrome and arteriovenous malformations. As such, we have included a discussion of the study's limitations and the need for future research in the conclusion section of this article.

Point 2: We appreciate your comments on the need for better organization and structure in the article, and we agree that this can enhance the readability and overall quality of the paper.

Based on your suggestions, we have revised the manuscript and have structured it into the following sections:

  1. Introduction
  2. Case Presentation

2.1. Diagnostic procedures

2.2. Treatment

  1. Discussion
  2. Conclusions

Point 3: We believe that this revised structure will make the article more accessible and easier to navigate for readers. We have also taken into account your comment on the need to carefully consider the specific case and have provided a detailed discussion on its implications and limitations.